

# Evaluation of urinary density as a biomarker for the diagnosis of acute heart failure

Mustafa Ahmet Akçalı[1], Semih Çınar[2], Kemal Abid Tekin[3], Recep Murat Mert[1], Sena Erduhan[4], Ertuğ Dinçer[5], Yusuf Altunöz[6], Arif Aksu[7] and Esra Akçalı[8]

[1] Department of Emergency Medicine, Ministry of Health Dogubayazit Dr Yasar Eryilmaz State Hospital, Ağrı, Turkey
[2] Department of Emergency Medicine, Tekirdag Dr. Ismail Fehmi Cumalioglu City Hospital, Tekirdağ, Turkey
[3] Department of Cardiology, Ministry of Health Dogubayazit Dr Yasar Eryilmaz State Hospital, Ağrı, Turkey
[4] Department of Medical Biochemistry, Dogubayazit Dr Yasar Eryilmaz State Hospital, Ağrı, Turkey
[5] Department of Emergency Medicine, Ministry of Health Merzifon Kara Mustafa Pasa State Hospital, Amasya, Turkey
[6] Department of Emergency Medicine, Sincan Nafiz Korez Education and Research Hospital, Ankara, Turkey
[7] Department of Emergency Medicine, Health Science University Adana City Research and Training Hospital, Adana, Turkey
[8] Department of Nephrology, Ministry of Health Tarsus State Hospital, Mersin, Turkey

Corresponding author
Mustafa Ahmet Akçalı,
ahmetakcali3@gmail.com

## ABSTRACT

**Background:** Heart failure (HF) has become a public healthcare concern with significant costs to countries because of the aging world population. Acute heart failure (AHF) is a common condition faced frequently in emergency departments, and patients often present to hospitals with complaints of breathlessness. The patient must be evaluated with anamnesis, physical examination, blood, and imaging results to diagnose AHF. Brain natriuretic peptide (BNP) is a widely accepted biomarker for the diagnosis of HF.
**Methods:** The files of the patients who applied to the emergency department with complaints of breathlessness were scanned, and BNP and urinary density (UD) levels were evaluated for the diagnosis of HF in patients.
**Results:** The results support that BNP is an effective biomarker in AHF, as is widely accepted. When the correlation between BNP and UD measurements was examined in the present study, a negative correlation was detected between the parameters. The results also suggested that low UD values may help diagnose AHF.
**Conclusion:** If similar results are obtained in prospective multicenter studies with the participation of more patients, UD value can be used as a biomarker for the diagnosis of AHF.

## INTRODUCTION

Heart failure (HF) is a common syndrome faced frequently in emergency departments as a significant cause of morbidity and mortality worldwide (*Castiglione et al., 2022*). European Society of Cardiology (ESC) defined HF as a clinical syndrome that may be accompanied by symptoms such as pulmonary crackles, elevated jugular venous pressure, and peripheral edema and consists of cardinal symptoms such as ankle swelling, breathlessness, and fatigue (*McDonagh et al., 2021*). Acute heart failure (AHF) is defined as a condition that causes the patient to apply to the emergency department or unplanned hospitalization as a result of the onset and aggravation of signs and/or symptoms of HF (*McDonagh et al., 2021*).

More than 1,000,000 patients are hospitalized due to HF each year in the United States of America (USA), and similar rates are reported in countries represented by the ESC (*Mebazaa et al., 2010*). Heart failure accounts for 5.8% of all Medicare charges and 3.6% of the total national hospital bills in the USA (*Heart Failure Executive Committee et al., 2008*). Slightly more than half of HF patients are women (*McDonagh et al., 2021*). The incidence of HF has been increasing in the last few years, and the elderly make up a large portion of this rate (*Kaluzna-Oleksy & Jankowska, 2019*). In Europe, although the prevalence of HF is 1–2% in adults, this increases to over 10% for those aged 70 and over (*McDonagh et al., 2021*). In Turkey, the prevalence of HF is 2.9% (*Değertekin et al., 2012*). Heart failure has become an important social and public healthcare concern with the significant aging of the population (*Mitsuaki, 2019*). The 5-year mortality rate after HF is diagnosed is reported to be over 50% in previous studies (*McDonagh et al., 2021*).

Symptoms and findings alone will not be adequate to diagnose HF. The anamnesis, electrocardiography (ECG), chest radiography, electrolyte levels, liver/kidney function test values, troponin, arterial blood gas, complete blood count results, Echocardiography (ECO) results and B-type natriuretic peptide (BNP) or N-terminal pro-evaluation together with B-type natriuretic peptide (NT-proBNP) levels of patients are recommended to be checked in diagnosing HF (*McDonagh et al., 2021*). The most common symptom of HF is breathlessness (*Mebazaa et al., 2010*; *Heart Failure Executive Committee et al., 2008*; *Mueller et al., 2019*; *Martindale et al., 2016*). The evaluation of acute dyspnea will be improved by adding the BNP test to clinical evaluation in the emergency department. Previous studies suggest using BNP as a diagnostic test in patients with breathlessness and other symptoms suggestive of HF (*Castiglione et al., 2022*; *McDonagh et al., 2021*; *Mueller et al., 2019*; *Daniels, Mills & Mueller, 2022*; *Núñez-Marín et al., 2023*; *Januzzi et al., 2018*; *Roberts et al., 2015*; *McCullough et al., 2002*). Although the diagnosis of HF can be clarified in approximately 56% of patients presenting to emergency departments with breathlessness through anamnesis, physical examination, chest radiograph, and ECG; however 44% remains unclear. When the measurement of natriuretic peptide (NP) values is added to the existing modalities, the accuracy of the diagnosis increases to 71–86% (*Martindale et al., 2016*). Many novel biomarkers, including molecular and gene studies, continue to be investigated to assist in the diagnosis of HF (*Sheng et al., 2022*). Urine density (UD), which is considered to be a candidate as a biomarker, is defined as the weight of the urine solution compared to an equal volume of distilled water (*Travers et al., 2023*).

 

The purpose of the present study was to evaluate the usability of UD value as a novel biomarker by comparing it with the widely accepted BNP value, which we think may help diagnose HF in patients presenting to emergency departments with breathlessness. Thus, we aimed to expand the diagnostic potential of BNP over UD. By examining this relationship, our study aims to supplement BNP with a new biomarker and thus improve diagnostic accuracy.

## MATERIALS AND METHODS

The approval of Ağrı İbrahim Çeçen University Faculty of Medicine Ethics Committee was obtained for the study (E-95531838-050.99-89760). The present study was conducted in the emergency department of a secondary care hospital in eastern Turkey, where approximately 60,000 adult patients are admitted on an annual basis. In this cross-sectional study, informed consent was not obtained from the participants and it was conducted as a retrospective file review after ethics committee approval was obtained.

Patients admitted to the emergency department with a complaint of breathlessness between 01.01.2022 and 15.12.2023, and whose ECG data, chest radiography, electrolyte levels, liver/kidney function test values, troponin, arterial blood gas, full blood count were recorded on the day of admission in the Hospital Information System (HIS) and those who had blood count results, complete urinalysis (CUA) results, BNP level, and whose ECO was performed and recorded by a cardiologist were included in the study. Patients who had a disease and/or symptom such as diabetes insipidus (DI), hematuria, pyuria, myoglobinuria, hemoglobinuria, renal failure, sepsis, severe burns, acute stroke, and obese patients were excluded from the study.

The sample size was calculated assuming a 95% confidence level and a 5% margin of error for a sensitivity of 91% measured at a cut-off value of BNP 100 from a previous study (*Hill et al., 2014*). Daniel's formula was used in the calculation as shown below (*Daniel, 2009*).

$$n = Z^2 \cdot p \cdot (1 - p)/d^2$$

where n = minimum sample size. Z = The table value for standard normal deviation corresponding to 95% significance level (= 1.96). P = prevalence of characteristic being estimated. d = margin of error, set at 0.05.

$$n = (1.96)^2 \cdot 0.91 \cdot (1 - 0.91)/(0.05)^2 = 126.$$

After adding 10% for the probability of having incomplete data, the sample size was found to be 140. A convenient sampling method was used to extract all the available medical records of the eligible patients from January 1, 2022, to December 15, 2023.

Since low UD values are expected in patients with DI, patients with a previous diagnosis of DI and patients with clinical findings compatible with DI, especially extreme thirst and severe dehydration in addition to dilute urine, were excluded (*Mutter et al., 2021*). Since high UD was expected in patients with hematuria, pyuria, myoglobinuria, and hemoglobinuria, these patients were excluded from the study (*Mutter et al., 2021*; *Simerville, Maxted & Pahira, 2005*). Since BNP values were expected to be high in patients

diagnosed with renal failure (GFR < 60), sepsis, severe burns, and acute stroke, and as BNP values were expected to be low in obese patients (BMI > 30), these patients were excluded from the study (*McDonagh et al., 2021*; *Mueller et al., 2019*; *Daniels, Mills & Mueller, 2022*; *Chen-Tournoux et al., 2010*; *Samad, Malempati & Restini, 2023*).

All patient files included in the study were examined by a team that consisted of cardiology, pulmonology, internal medicine, and emergency medicine specialists and were divided into two groups as those who were diagnosed with AHF and those without. Receiver operating characteristic (ROC) analysis was performed and cut-off values were determined to evaluate the power of BNP and UD measurements in diagnosing AHF. Also, the BNP cut-off value, which is widely accepted in the literature, was accepted as 100 pg/ml, and the power of BNP measurement was evaluated to predict the diagnosis of AHF (*Castiglione et al., 2022*; *McCullough et al., 2002*).

The data of the study were analyzed in the IBM SPSS Statistics for Windows, Version 25.0 (Released 2011; IBM Corp., Armonk, NY, USA). Descriptive statistics were given as numbers and percentages for categorical variables. The suitability of numerical variables for normal distribution was evaluated with the Shapiro-Wilk test. The variables that complied with normal distribution were expressed as mean ± standard deviation (SD), and variables that did not comply with normal distribution were expressed as median and minimum-maximum values (min-max). The chi-square test was used for the analysis of the categorical variables, the Mann-Whitney U-test was used for the analysis of the numerical variables that did not comply with normal distribution in independent groups. The Student *t*-Test was used for the analysis of the numerical variables that did not comply with normal distribution in independent groups. The relationships between the numerical variables that did not comply with normal distribution were examined with the Spearman Correlation Analysis. A cut-off value was determined by performing ROC Analysis to calculate the sensitivity and specificity, positive and negative likelihood ratios, and positive and negative predictive values of BNP and UD measurements in predicting the diagnosis of AHF. The statistical significance level was taken as $p < 0.05$.

## RESULTS

The average age of 140 patients who were included in the study was found to be 70.7 ± 10.0, and 80 patients were female. The demographic data, AHF status, laboratory, and ECO findings of the patients are given in Table 1.

The relationship between the AHF status of the patients and their demographic data, laboratory parameters, and ECO findings is summarized in Table 2. No statistical differences were detected between the AHF status, age, and gender ($p > 0.05$), and a statistically significant difference was found between laboratory and ECO findings ($p < 0.05$).

When the correlation between BNP and UD measurements was evaluated, a moderate and negative correlation was detected between the parameters (rho = −0.530, $p < 0.05$).

In the ROC analysis that was used to evaluate the power of the BNP value in predicting the presence of AHF, the BNP cut-off value was found to be 106.0. When 106.0 was used as the BNP cut-off value, the sensitivity of BNP measurement was found to be 88.24% in

**Table 1 The demographic data of the patients, acute heart failure status, laboratory and echocardiography findings.**

| Age | Mean ± SD |
|---|---|
| Year | 72.6 ± 9.1 |
| **Gender** | ***n* (%)** |
| Female | 80 (57.1%) |
| Male | 60 (42.9%) |
| **Heart failure** | ***n* (%)** |
| Yes | 68 (48.6%) |
| No | 72 (51.4%) |
| **Laboratory and echocardiography parameters** | **Median (min–max)** |
| BNP (pg/mL) | 113.0 (10.0–2,700.0) |
| Urine density | 1,011 (1,000–1,120) |
| Ejection fraction (%) | 60 (20–60) |

Note:
BNP, Brain natriuretic peptide.

**Table 2 The relationship between acute heart failure status and demographic data, laboratory parameters and echocardiography findings of the patients.**

| | Heart failure | No heart failure | *p*-value |
|---|---|---|---|
| **Age (years)** | 74.0 ± 9.6 | 71.3 ± 8.5 | 0.080* |
| **Gender** | | | |
| Female, *n* (%) | 36 (52.9%) | 44 (61.1%) | 0.329# |
| Male, *n* (%) | 32 (47.1%) | 28 (38.9%) | |
| **Laboratory and echocardiography parameters** | | | |
| BNP (pg/mL) | 385.0 (19.0–2,700.0) | 37.5 (10.0–452.0) | <0.000** |
| Urine density | 1.007 (1.000–1.018) | 1.019 (1.004–1.120) | <0.000** |
| Ejection fraction (%) | 55 (20–60) | 60 (45–60) | 0.002** |

Notes:
BNP, Brain natriuretic peptide.
* *p*-value for the student **t**-test.
# *p*-value for the chi-square test.
** *p*-value for the MWU test.

predicting the presence of AHF (95% CI [78.13–94.78%]), with a specificity of 84.72% (95% CI [74.31–92.12%]), a positive likelihood ratio of 5.78 (95% CI [3.33–10.03]), a negative likelihood ratio of 0.14 (95% CI [0.07–0.27]), a positive predictive value of 84.51 (95% CI [75.87–90.44%]), a negative predictive value of 88.41% (95% CI [79.79–93.64%]), and an accuracy rate of 86.43% (95% CI [79.62–91.63%]) (AUC = 0.914, 95% CI [0.863–0.964], $p < 0.05$).

The results calculated using 106.0 as the BNP cut-off value are similar to the results calculated using 100.0 as the BNP cut-off value.

In the receiver operating characteristic (ROC) analysis that was used to evaluate the power of UD value in predicting the presence of AHF, the cut-off value of UD was found to be 1.013. When 1.013 was used as the UD cut-off value, the sensitivity of UD measurement was 91.18% in predicting the presence of AHF (95% CI [81.78–96.69%]), with a specificity

**Table 3 The distribution of the patients according to the presence of acute heart failure, BNP ≤ 106 pg/mL, BNP ≤ 100 pg/mL, and urine density ≤ 1.013.**

|  | Have heart failure | No heart failure | *p*-value |
|---|---|---|---|
| **BNP** |  |  |  |
| BNP ≤ 106 pg/mL | 8 (5.7%) | 61 (43.6%) | <0.000* |
| BNP > 106 pg/mL | 60 (42.9%) | 11 (7.9%) |  |
| **BNP** |  |  |  |
| BNP ≤ 100 pg/mL | 8 (5.7%) | 61 (43.6%) | <0.000* |
| BNP >100 pg/mL | 60 (42.9%) | 11 (7.9%) |  |
| **Urine density** |  |  |  |
| Urine density ≤ 1.013 | 62 (44.3%) | 13 (9.3%) | <0.000* |
| Urine density > 1.013 | 6 (4.3%) | 59 (42.1%) |  |

Notes:
BNP, Brain natriuretic peptide.
* *p*-value for chi-square test.

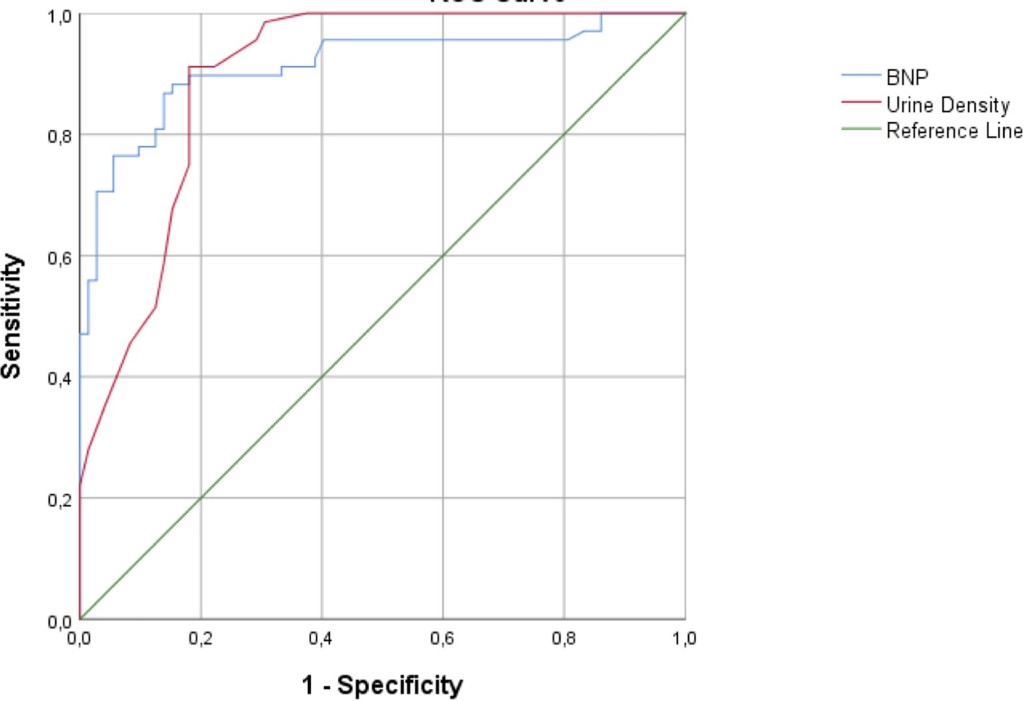

**Figure 1 The comparison of the BNP and urine density measurements with ROC curves for predicting the presence of acute heart failure.** BNP, Brain natriuretic peptide.

of 80.82% (95% CI [69.92–89.10%]), a positive likelihood ratio of 4.75 (95% CI [2.95–7.65]), a negative likelihood ratio of 0.11 (95% CI [0.05–0.24]), a positive predictive value of 81.58 (95% CI [73.33–87.70%]), a negative predictive value of 90.77% (95% CI [81.96–95.51%]), and an accuracy rate of 85.82% (95% CI [78.95–91.12%]) (AUC = 0.896, 95% CI [0.843–0.949], *p* < 0.05). The distribution of the patients according to the presence of AHF and BNP ≤ 106 pg/mL, BNP ≤ 100 pg/mL, and UD ≤ 1.013 is given in Table 3. The

comparison of BNP and UD measurements with ROC curves for predicting the presence of AHF is given in Fig. 1.

Also, when HF patients were evaluated according to left ventricular ejection fraction (LVEF), HF with preserved ejection fraction (HFpEF) (LVEF ≥ 50%), HF with mildly reduced ejection fraction (HFmrEF) (LVEF 41–49%) and HF with reduced ejection fraction (HFrEF) (LVEF ≤ 40%) rates were found to be HFpEF 63.2%, HFmrEF 16.2%, and HFrEF 20.6%, respectively. This rate is similar to the results of the existing studies (*McDonagh et al., 2021*).

## DISCUSSION

The number of female patients who were diagnosed with HF was higher than the number of males in previous studies, and the incidence of HF increases with age, exceeding 10% for those who are aged 70 and over (*McDonagh et al., 2021*; *Kaluzna-Oleksy & Jankowska, 2019*). Similarly, the mean age of HF patients was 74.0 ± 9.6 in the present study and the number of female patients was higher than that of males. The probable reason for this might be that the average lifespan of women is longer than the average lifespan of men.

*McCullough et al. (2002)* conducted a study in seven centers and reported that the sensitivity of BNP above 100 pg/ml in diagnosing HF in patients presenting with acute dyspnea was 90%, and the specificity was 73%. The area under the ROC curve (AUC) was 0.90 (95% CI [0.88–0.91]) for the diagnosis of HF at the cut-off point of 100 pg/ml ($p < 0.0001$). In another study conducted by *Castiglione et al. (2022)*, the AUC of BNP for the diagnosis of HF at the cut-off point of 100 pg/ml was 0.91, with 90% sensitivity, 76% specificity, and 83% accuracy in 1,586 patients admitted to the emergency department because of new-onset dyspnea. Also, *Hill et al. (2014)* examined the sensitivity and specificity of BNP in diagnosing HF in 51 articles, and the results were found to be 91% and 80%, respectively. In the present study, the BNP cut-off value was 106.0 in the ROC analysis that was used to evaluate the power of the BNP for predicting the presence of AHF. When the BNP cut-off value was calculated as 100 and 106 pg/ml, the results obtained were similar to those of the existing studies.

The results could not be compared with similar studies because of the limitations of studies regarding the use of UD value as a biomarker for the diagnosis of AHF. However, a decrease in urinary urea (73%), chloride (5.4%), sodium (5.1%), potassium (2.4%), phosphate (2.0%), uric acid (1.7%) and sulfate (1.3%) particles in urine reduces urinary density (*Pradella, Dorizzi & Rigolin, 1988*). *Nawrocka-Millward et al. (2024)* found that low urinary chloride levels were associated with low urinary urea, sodium and potassium, and patients with low urinary chlorine concentrations exhibited higher HF burden or more advanced disease stage and worse in-hospital prognosis. Patients with low urinary sodium levels exhibited similar features in AHF (*Honda et al., 2018*; *Martens et al., 2019*). These findings support the potential of UD as a biomarker for the diagnosis of AHF. Although the density of distilled pure water is 1 g/ml, the density of urine is generally measured between 1.013 and 1.029 g/ml in healthy individuals (*Travers et al., 2023*). The urinary density cut-off value was found to be 1.013 in the ROC analysis performed to evaluate the power of the UD value in predicting the presence of AHF in the present study. When 1.013

was used as the urine density cut-off value, the sensitivity of UD measurement was 91.18% for predicting the presence of AHF, the specificity was 80.82%, the positive likelihood ratio was 4.75, the negative likelihood ratio was 0.11, and the AUC value was 0.896 and 95%. We think that UD, which has a high sensitivity and specificity, of 1.013 and below, can be used to predict the diagnosis of AHF.

When the correlation between BNP and UD measurements was evaluated, a moderate and negative correlation was detected between the parameters ($p < 0.05$).

Our study evaluates the relationship between UD and BNP and suggests an additional biomarker for the diagnosis of AHF. Our findings suggest that UD can be used in the diagnosis of AHF.

There are some limitations in our study. These include: the study was conducted in one single center. More data can be obtained through prospective multicenter studies including more patients. Another limitation of the study was that patients with additional diseases that would significantly change the BNP values or UD values were not included in the study.

## CONCLUSIONS

In conclusion, BNP value is used as a widely accepted and important biomarker in the diagnosis and exclusion of AHF in patients presenting to the emergency department with complaints of breathlessness. The results supporting this were obtained in the present study.

In our study, UD value and BNP results of AHF patients showed a significant negative correlation. In particular, a UD value of ≤1.013 strongly supports the diagnosis of AHF.

If similar rates are found in multicenter studies to be conducted with more patients in the future, the use of UD as a biomarker in the diagnosis of AHF, which is a cheap, rapid and easily accessible diagnostic tool, may make a significant contribution to both increasing diagnostic accuracy and providing economic contribution to the healthcare system.

### Funding
The authors received no funding for this work.

### Competing Interests
The authors declare that they have no competing interests.

### Author Contributions
- Mustafa Ahmet Akçalı conceived and designed the experiments, performed the experiments, prepared figures and/or tables, authored or reviewed drafts of the article, and approved the final draft.
- Semih Çınar analyzed the data, authored or reviewed drafts of the article, and approved the final draft.

- Kemal Abid Tekin conceived and designed the experiments, authored or reviewed drafts of the article, and approved the final draft.
- Recep Murat Mert performed the experiments, authored or reviewed drafts of the article, and approved the final draft.
- Sena Erduhan conceived and designed the experiments, authored or reviewed drafts of the article, and approved the final draft.
- Ertuğ Dinçer analyzed the data, prepared figures and/or tables, authored or reviewed drafts of the article, and approved the final draft.
- Yusuf Altunöz analyzed the data, authored or reviewed drafts of the article, and approved the final draft.
- Arif Aksu analyzed the data, authored or reviewed drafts of the article, and approved the final draft.
- Esra Akçalı conceived and designed the experiments, authored or reviewed drafts of the article, and approved the final draft.

### Human Ethics

The following information was supplied relating to ethical approvals (*i.e.*, approving body and any reference numbers):

Ağrı İbrahim Çeçen University Rectorate Legal Counseling Office (Ethical Application Ref: E-95531838-050.99-89760).

### Data Availability

The raw measurements are available in the Supplemental File.

### Supplemental Information

Supplemental information for this article can be found online at http://dx.doi.org/10.7717/peerj.18836#supplemental-information.

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
