# Peer review of "Evaluation of urinary density as a biomarker for the diagnosis of acute heart failure"

_PeerJ, doi:10.7717/peerj.18836_

## Round 0.1 · original submission · Major Revisions

Please revise your manuscript according to the reviewers' comments.
Yours,
Yoshi
Prof. Yoshinori Marunaka, M.D., Ph.D.

·

Basic reporting

The manuscript is well-written with clear language, and it is well-structured from the background till the conclusion.
Here are my comments and suggestions in the introduction part of your manuscript.
1. I suggest you to mention the prevalence of acute heart failure in the study area or at national level in Turkey.
2. Define the abbreviation ''AC'' in the 3rd paragraph
3. I strongly advise you to highlight the role of urine density in the diagnosis of acute heart failure based on previous research findings.

Experimental design

The research is within the aim and scope of the journal.
Research questions are relevant and meaningful.
Please consider my suggestions and comments regarding materials and methods as mentioned below
1. I suggest you to provide an explanation how diabetes insipidus was excluded (with clinical evidences or lab testing).
2. It would be better if the reasons why stroke and obese patients are excluded in the methods section than in the discussion part.
3. The reason to use a cut-off value of BNP (100pg/mL) should be made clear and cited with a reference.
4. Define the abbreviation ''ROC''.
5. I suggest you to show how samples size was calculated and to mention the sampling technique used in your study.

Validity of the findings

The significance of the literature is clearly stated.
The underlying data are statistically sound.
Conclusions are well stated.
But I do have some comments.
1. When you state the results, you should not mention the statistical tests unless mandatory, as they already mentioned as foot notes under the tables.
2. Results with cut-off values of BNP with 100 and 106 pg/mL are similar but mentioned for both. Hence, rewite this section with no redundancy.
3. The percentages of patients based on ejection fraction should be mentioned in the results section before it is mentioned in the discussion part.
4. Use appropriate terminology for ejection fraction between 41% and 49%; replace intermediate with mildy reduced.

Additional comments

1. Define every abbreviation in its first appearance in the manuscript and use abbreviations consistently once they are defined. For instance, AHF and acute heart failure have been used alternatively and this should be corrected.
2. Put firm conclusions and strong recommendations in your conclusion and don't start such statements with ''we think'' and ''we believe''.

·

Basic reporting

- Overall this article sufficient for publication
- Minor revision needed to make this article stand out especially in method explanation in abstract and manuscript
- Articles structure, figures, tables and raw data sufficient to support the manuscript
- Relevant issue and hypotheses of this articles already contained novelty to expand the benefit of BNP but please explain it further to make it clearer and brighter

Experimental design

- Aims and Scope are sufficient to support general description and originality of the manuscript
- Research question relevant and meaningful
- Investigation performed in sufficient way and according to ethical standard
- Methods needs a little bit improvement in explanation to make it perfect

Validity of the findings

- Novelty already contained in the manuscript but needs to be improved in explanation to give more insightful benefit for the reader
- Data sufficient to support manuscript
- The benefit and limitation needs to be stated in discussion
- Conclusions needs to be explained in the specific way to elevate the meaning of the paragraph

Additional comments

This is a very meaningful study, good job for the research team, best regards

---

## Round 0.2 · accepted · Accept

In my opinion, your article is now Acceptable. Congratulations.
Yours,
Yoshi
Prof. Yoshinori Marunaka, M.D., Ph.D.